# Effects of Dietary Chromium Picolinate on Gut Microbiota, Gastrointestinal Peptides, Glucose Homeostasis, and Performance of Heat-Stressed Broilers

**DOI:** 10.3390/ani12070844

**Published:** 2022-03-27

**Authors:** Guangju Wang, Xiumei Li, Ying Zhou, Jinghai Feng, Minhong Zhang

**Affiliations:** State Key Laboratory of Animal Nutrition, Institute of Animal Sciences, Chinese Academy of Agricultural Sciences, Beijing 100193, China; 82101185163@caas.cn (G.W.); llxiumei93@163.com (X.L.); 15624955881@163.com (Y.Z.); fjh6289@126.com (J.F.)

**Keywords:** heat stress, glucose homeostasis, microbiota, gastrointestinal peptide, organic chromium

## Abstract

**Simple Summary:**

High ambient temperature is a common environmental issue in poultry breeding. It not only seriously damaged the growth performance of broilers, but also had adverse effects on intestinal microbial composition, gastrointestinal peptide, and glucose homeostasis. Trivalent chromium has the ability to improve the absorption and utilization of glucose. To date, the impacts of trivalent chromium on the above indexes remain only partially clear. This research discussed the influence of chromium supplementation on performance, microbiota, gastrointestinal peptides, and glucose homeostasis in heat-stressed broilers.

**Abstract:**

The current research was devoted to evaluating the effects on gut microbiota, gastrointestinal peptides, and glucose homeostasis of chromium picolinate applied to heat-stressed broilers. In a 14 d experiment, 220 28-day-old AA broilers were randomly assigned into one thermal-neutral and three high-temperature groups dietary-supplemented with 0, 0.4, or 0.8 mg/kg of chromium as chromium picolinate. The temperature for the thermal-neutral group was set at 21 °C, while that for the other three groups (high temperature) was set at 31 °C. The results showed that the average daily gain and average daily feed intake of the 0.4 mg/kg chromium-supplemented group significantly increased compared with the high-temperature groups (*p* < 0.05). The content of cholecystokinin in the 0.4 mg/kg group significantly decreased, and the gastric inhibitory polypeptide level was significantly elevated in jejunum (*p* < 0.05). The cecal microbiota of heat-stressed broilers was substantially different from that of the thermal-neutral group. After diet-supplemented chromium, compared to the high-temperature groups, the 0.4 mg/kg chromium supplemented group was characterized by a reduction of Actinobacteriota and Proteobacteria at the phylum level. The Bacilli were elevated, while proportions of Coriobacteria and Gammaproteobacteria were reduced significantly at the class level. The proportions of Lactobacillaceae, Christensenellaceae, and Erysipelotrichaceae were elevated significantly, while that of Clostridiaceae was reduced significantly at the family level. The proportion of Turicibacter was elevated significantly and the proportions of Olsenella and Ruminococcus were reduced significantly at the genus level (*p* < 0.05). Compared to the high-temperature groups, in the 0.4 mg/kg chromium-supplemented group, the insulin concentration and insulin resistance index were reduced (*p* < 0.05), and sodium-glucose transporter 1 expression was up-regulated in jejunum (*p* < 0.05). Performance, microbiota, gastrointestinal peptides, or serum parameters of the 0.8 mg/kg group were almost unaffected by chromium compared with the high-temperature groups. In conclusion, diet supplemented with 0.4 mg/kg Cr improved performance, insulin resistance and sodium-glucose transporter 1 expression and altered gut microflora structure and secretion of gastrointestinal peptides, thus showing that supplementation with chromium is beneficial to maintain glucose homeostasis and alleviate heat stress.

## 1. Introduction

High ambient temperature is a primary problem in broiler breeding, since it has a negative impact on performance and other traits that affect profits [1]. Evidence in several studies indicates that glucose homeostasis can be influenced under heat stress in animals [2,3], including changes in blood glucose, insulin, and NEFA [4,5,6]. More seriously, when target tissues cannot respond well to insulin and the insulin level becomes higher than the level of glucose in the circulation, animals develop insulin resistance as a result. A previous study observed a state of insulin resistance in human hyperthermia [7]. Our previous research found that heat stress caused insulin resistance in broilers [8]. High temperature leads to reduced glucose absorption efficiency. Recent evidence suggests that excessive temperature affects glucose transporter expression in broilers [9]. Furthermore, excessive temperature impacts the composition of the gut microbial community and gastrointestinal peptides. Our previous research suggested that high temperature significantly alters the proportion of bacteria and abundance of metabolites and that the concentrations of several gastrointestinal peptides changed owing to excessive temperature [8].

Chromium is a trace nutrient and an important active component of the glucose tolerance factor, which regulates the metabolism of carbohydrates, fat, and protein [10]. It has been confirmed that heat stress will increase the excretion of chromium in broilers, and that chromium deficiency will lead to an imbalance in glucose homeostasis, as well as blunt glucose response [11,12]. Numerous research studies have indicated that dietary trivalent chromium can alleviate the adverse impact caused by heat stress [13,14,15]. Trivalent chromium could improve insulin sensitivity and glucose utilization in animals and humans who suffer from metabolic diseases such as type 2 diabetes and obesity [16,17]. In several studies, the glucose transporters in heat-stressed poultry could be improved by supplementing their diets with organic chromium, which is conducive to maintaining glucose homeostasis [18,19,20]. However, there has been no report about the influence of chromium on glucose homeostasis involved in the microflora in heat-stressed chickens.

Thus, the objective of this paper was to assess the influence of chromium picolinate supplementation on gut microbiota, glucose transporters, gastrointestinal peptides, glucose homeostasis, and performance in broilers raised under heat stress, then, to explore the effect of chromium picolinate on alleviating heat stress through the microbiota and provide a new regulatory target for alleviating heat stress in broilers.

## 2. Materials and Methods

### 2.1. Full Name and Description of Abbreviations

For abbreviations, see the abbreviations list presented in Table 1.

### 2.2. Animals

A total of 270 male 1-day-old AA broilers were accessed from a local hatchery and reared in breeding cages (three layers, 8400 cm^2^ per layer) according to AA broiler breeding management requirements. All birds were provided access to water and feed ad libitum. A total of 220 birds were selected according to average weight, then randomly assigned to four groups with five replicates (11 chickens/pen) according to individual body weight. The treatment group design is shown in Table 2. The adaptation period was entered at the age of 21 days, and the trial began at the age of 28 days and lasted from 14 days to 42 days. Ambient temperature and humidity control were adjusted by real-time temperature and humidity control chambers developed by our Institute. The chamber with one door has good thermal insulation performance. The average body weight of the four groups of broilers was similar. The temperature in the chambers of each group was constant, and the humidity was maintained at 60% during the experiment. The broilers were fed on a crumble diet that was selected to meet the NRC nutrient recommendations (1994, Table 3). Birds were provided with continuous light (20 lux, 18 h light, 6 h dark) throughout the experimental period. To reduce stress, the number of times in and out of the chamber was minimized. Ethical approval was obtained based on the animal welfare and ethics checklist of the institute of animal science of the Chinese academy of agricultural sciences (ethics code number: IAS 2021-75).

### 2.3. Samples and Data Collection

A scale with accuracy to 0.01 g was used to measure the body weight of birds in each replicate. The ADFI, ADG, and FCR for each replicate were calculated at the end of the trial. One bird was randomly selected out of the 11 birds in each replicate for further sampling; serum was collected and stored in a refrigerator for determination of the levels of ghrelin, CCK, GIP, and other parameters. When serum sample collection was completed, the birds with blood samples taken were immediately humane-sacrificed and the cecal contents, hypothalamus, and intestinal mucosa were obtained. The samples were stored in a −80 °C refrigerator. Another chicken was randomly selected from each replicate after 12 h fasting for insulin and insulin resistance index determination.

### 2.4. Determination of Gastrointestinal Peptides

ELISA was used to measure the contents of CCK, ghrelin, and GIP in the hypothalamus, intestinal mucosa, and serum. The intestinal mucosa was ground into a powdery form, diluted with PBS (1:9), then centrifuged at 2000–3000 rpm for 20 min; the supernatant was collected for further determination. Standard wells, sample wells, and blank wells were set, then 50 μL serum samples were added into each sample well. The diluent and enzyme label reagent were successively added and incubated at 37 °C for 60 min. A washing solution was used to wash the ELISA plate, then a color reagent and stop solution were added, and eventually the enzyme-labeled instrument (Bio-tek, Gen 5 1.10) was used to determine the absorbance of each well. The levels of peptides were calculated according to the standard curve.

### 2.5. Analysis of the Cecal Microbiota

DNA of microbiota was extracted from cecal content samples using the E.Z.N.A.^®^ soil DNA Kit (Omega Bio-tek, Norcross, GA, USA). The hypervariable region V3-V4 of the 16S rRNA gene was amplified with primer pairs 338F (5′-ACTCCTACGGGAGGCAGCAG-3′) and 806R (5′-GGACTACHVGGGTWTC-TAAT-3′). The PCR amplification procedure was as follows: 95 °C for 3 min, 27 cycles at 95 °C for 30 s, 55 °C for 30 s, 72 °C for 45 s, 72 °C for 10 min, ending at 10 °C. The PCR product was extracted from 2% agarose gel and purified. Purified amplicons were pooled in equimolar ratios and paired-end sequenced on an Il-lumina MiSeq PE300 platform/NovaSeq PE250 platform (Illumina, San Diego, CA, USA) according to the standard protocols.

### 2.6. Determination of SCFAs

One gram of digesta from the cecum samples was weighed accurately. The samples after dilution with ultra-pure water were subjected to short-chain acid analysis by gas chromatography. Chromatographic conditions were as follows. Column: db-ffap (30 m × 250 μm × 0.25 μm), flow rate: 0.8 mL/min, auxiliary gas: high purity hydrogen (99.999%), the temperature of detector FID: 280 °C, temperature of injection port: 250 °C, split ratio: 50:1, injection volume: 1 μL, temperature programming: 60 °C→220 °C (20 °C/min).

### 2.7. Determination of Glucose Transporter Gene Expression

The expression levels of SGLT1 and GLUT2 in the jejunum were measured using qPCR. The general total RNA extraction kit (Hooseen biology, Beijing, China) was used to extract RNA from tissue samples, and the experimental operation was carried out according to the product instructions. Real-time PCR was conducted using the Line Gene 9620 Real-time PCR System (Bioer, Hangzhou, China). Primer-set sequences are provided in Table 4. Real-time PCR reactions were conducted at 95 °C for 10 min, followed by 40 cycles of 95 °C for 20 s, 55 °C for 20 s and 72 °C for 20 s. The mRNA levels were normalized to glyceraldehyde-3-phosphate dehydrogenase levels (ΔCt). The relative gene expression amount of each sample was calculated according to the formula for relative quantities: RQ = 2^−^^ΔΔCt^.

### 2.8. Determination of Glucose Homeostasis and Other Serum Parameters

Glucose homeostasis determination was performed by insulin resistance index, using the formula insulin resistance index = insulin/(22.5e-ln glucose) to calculate the insulin resistance index [21]. The levels of insulin, NEFA, TG, and TC were determined using the kits provided by Nanjing Jiancheng Bioengineering Institute (Jiangsu, China).

### 2.9. Statistical Analysis

All data were analyzed using GraphPad Prism 8 software (GraphPad Prism 8.3.0 for Mac OS, GraphPad Software, Inc., La Jolla, CA, USA). One-way ANOVA followed by post hoc (Bonferroni’s multiple comparison test) analysis was used for the comparison of four treatments. The confidence interval of 95% was considered, *p* < 0.05 means a significant difference, and values are presented as the mean.

## 3. Results

### 3.1. Effect of Chromium Supplementation on Performance of Heat-Stressed Broilers

The performance of broilers decreased significantly due to high ambient temperature (Table 5). The ADFI of the TN group was higher than those of the HT, 0.4 mg/kg and 0.8 mg/kg groups (*p* < 0.05), and the ADFI of the 0.4 mg/kg group was higher than those of the HT and 0.8 mg/kg groups (*p* < 0.05). High ambient temperature decreased the ADG (*p* < 0.05), while the ADG of the 0.4 mg/kg group was higher than that of the HT group, the FCRs of the HT, 0.4 mg/kg, and 0.8 mg/kg groups were elevated (*p* < 0.05), and there were no significant differences in FCR between the 0.4 mg/kg, 0.8 mg/kg, and HT groups.

### 3.2. Effects of Chromium Supplementation on Gastrointestinal Peptides

The effects of chromium supplementation on gastrointestinal peptides of the gut and hypothalamus in broilers under heat stress were measured, and the results are presented in Figure 1 and Figure 2. In the hypothalamus, the high ambient temperature increased the concentration of ghrelin (*p* < 0.05), but there was no significant difference between the two chromium supplementation groups and the TN group. The CCK concentrations in jejunum and serum were higher in the HT group than those in the TN group (*p* < 0.05), while the CCK concentration in the 0.4 mg/kg group was significantly lower than those in the HT and 0.8 mg/kg groups. There was no significant difference in jejunum and serum ghrelin among the four groups. In jejunum, compared with the TN group, the GIP content in the HT group was lower, while the GIP content in the 0.4 mg/kg group was significantly higher than those in the HT and 0.8 mg/kg groups. The CCK concentrations in jejunum and serum in the 0.4 mg/kg supplemented group were significantly lower than that in the HT group (*p* < 0.05). The CCK concentration in the hypothalamus, GIP in serum, and ghrelin in serum and jejunum in the 0.4 mg/kg supplemented group showed no difference from the corresponding values in the HT group.

### 3.3. Effect of Chromium Supplementation on Cecal Microbial Composition

We assessed the composition of the cecal flora of the TN, HT and 0.4 mg/kg supplemented groups of broilers. The structures of cecal microbial at the phylum, class, family, and genus levels are presented in Table 6. At the phylum level, the proportion of Actinobacteriota and Proteobacteria in the flora of the 0.4 mg/kg supplemented group decreased compared to the HT and TN groups (*p* < 0.05). At the class level, the proportion of Bacilli in the flora of the 0.4 mg/kg supplemented group was significantly increased, whereas Coriobacteriia and Gammaproteobacteria were significantly decreased compared with the HT and TN groups (*p* < 0.05). At the family level, the proportion of Christensenellaceae in the flora of the HT group decreased significantly compared with the TN group. Moreover, Clostridiaceae in the flora of the 0.4 mg/kg supplemented group was decreased, and Lactobacillaceae, Christensenellaceae, and Erysipelotrichaceae in the flora of the 0.4 mg/kg supplemented group were increased compared to the HT group (*p* < 0.05). At the genus level, Ruminococcus and Olsenella abundances in the flora of the 0.4 mg/kg supplemented group were significantly decreased, whereas the Turicibacter proportion in the flora of the of 0.4 mg/kg supplemented group was significantly increased compared with that of the HT group.

### 3.4. Effects of Chromium Supplementation on SCFA Concentrations

The concentrations of SCFAs in the cecal digesta of the four groups are presented in Table 7. Compared to the TN group, the concentration of acetate in the digesta of the HT group increased (*p* < 0.05). No significant differences in acetate, propionic acid and butyric acid concentrations were found in the 0.4 mg/kg or 0.8 mg/kg chromium supplemented groups compared with the HT group.

### 3.5. Effects of Chromium Supplemented on Glucose Homeostasis and Other Serum Parameters

We analyzed the levels of insulin, insulin resistance index, TG, TC, and NEFA, and the results are presented in Figure 3. Compared with the TN group, the levels of insulin, insulin resistance index, TG, and TC increased, and the level of NEFA decreased, in the HT group (*p* < 0.05). Compared with the HT group, the insulin level and insulin resistance index were significantly reduced (*p* < 0.05) in the 0.4 mg/kg chromium supplemented group, whereas there were no effects on TC, TG, and NEFA. However, there were no differences in insulin, insulin resistance index, TC, TG, and NEFA between the 0.8 mg/kg chromium supplemented group and the HT group.

### 3.6. Effects of Chromium Supplementation on the Level of Glucose Transporter Gene Expression

The effect of chromium supplementation on the level of glucose transporter gene expression is shown in Table 8. Compared with the TN group, the relative expression of the SGLT1 gene was decreased in the HT group. Compared with the HT group, the relative expression of the SGLT1 transporter gene in jejunum from the 0.4 mg/kg chromium supplemented group was significantly higher (*p* < 0.05). There were no significant changes in the relative expression of GLUT2 in liver (*p* > 0.05).

## 4. Discussion

Trivalent chromium has a positive effect on livestock under heat stress [13,22,23]. It has been reported that diets supplemented with 0.2 mg/kg Cr as Cr-Pic and Cr-His both significantly increased feed intake and body weight gain in heat-stressed broilers [24]. In the current study, the ADFI and ADG significantly increased in the 0.4 mg/kg supplemented group compared with the HT group, indicating that dietary supplementation with 0.4 mg/kg Cr as Cr-Pic can reduced the severity of impaired performance under heat stress. This corresponds with previous results indicating that Cr-Pic has a positive effect on growth performance in broilers under heat stress [25,26].

Several critical factors in glucose homeostasis include intestinal glucose uptake, peripheral tissue glucose uptake and utilization, and liver glucose output efficiency. Insulin is an important factor in controlling glucose homeostasis. Insulin resistance will lead to an imbalance in glucose homeostasis, abnormal changes in blood glucose, and even lead to metabolic diseases such as diabetes mellitus with hyperinsulinemia. Trivalent chromium has been proved to regulate glucose metabolism in coordination with insulin, which is conducive to maintaining glucose homeostasis [27]. In the current study, compared with the HT group, the serum insulin concentration and insulin resistance index of the 0.4 mg/kg group were significantly reduced. Chromium is a cofactor of insulin activity and an essential element for glucose utilization [28]. This may be due to increased insulin sensitivity resulting from chromium supplementation. Therefore, it indicated that 0.4 mg/kg Cr as Cr-Pic supplement was beneficial to relieve the insulin resistance caused by heat stress.

The intestinal microbiota is the largest and most complex micro-ecosystem in animals. The microflora are not only involved in a variety of metabolic pathways to regulate metabolism, but also play an important role as a mediator between the diet and the host. Undoubtedly, diet is a factor that influences the composition of the intestinal microbiota. In the 0.4 mg/kg chromium supplemented group of the current study, at the class level, the proportion of Coriobacteria decreased. In a previous study, it was mentioned that an increased proportion of Coriobacteria will increase the risk of insulin resistance and obesity [29]. At the family level, the proportions of Christensenellaceae, Clostridiaceae, and Erysipelotrichaceae increased; these bacteria were reported to be beneficial to normal glucose metabolism, alleviate insulin resistance, and lower the risk of obesity [30,31]. Our previous study showed that the abundance of Christensenellaceae was significantly reduced by heat stress, while the abundance of Christensenellaceae was significantly increased in the 0.4 mg/kg supplemented group [8], indicating that glucose metabolism in the chromium supplement group improved. At the genus level, the proportion of Ruminococcus decreased. Previous research suggests that microbiota can induce insulin resistance, and Ruminococcus is one of them [32]. Therefore, these alterations in microflora suggest that 0.4 mg/kg supplementation was beneficial to alleviate insulin resistance and improve glucose homeostasis.

The absorption of carbohydrates in the gut is carried out by glucose transporters. In poultry, the duodenum and jejunum are the main sites for the absorption and uptake of glucose [33]. SGLT1 transfers glucose to the epithelial cells of the small intestine through active transport. After intracellular transport, glucose is transported to all parts of the body through GLUT2 in the basement membrane in a diffusion-assisted manner [34]. GLUT2 is more sensitive when the body is hyperglycemic, which helps the liver to efficiently regulate the blood glucose level [34]. In addition, SGLT1 and GLUT2 play a vital role in incretin secretion [35]. The importance of the SGLT1 transporter in sodium-glucose homeostasis was confirmed by the disappearance of intestinal glucose uptake in SGLT1-deficient mice [36]. The current study suggested that the level of SGLT1 in the jejunum in the 0.4 mg/kg supplemented group was significantly up-regulated. These discoveries are consistent with those of Cemal Orhan et al., who cited a significant increase in SGLT1 expression in ileum of laying hens by supplementation with Cr-Pic [19]. Chromium factors are involved in glucose metabolism under heat stress; these forms of chromium can not be reabsorbed, leading to a lack of chromium and reduced efficiency of glucose absorption in the intestine. Thus, the improvement in SGLT1 expression could be due to the increase in chromium levels in the intestine from diet-supplemented chromium. Kashiwagi et al. also found decreased glucose transporters in insulin-resistant patients with type II diabetes [37]. Insulin resistance is associated with the efficiency of glucose absorption and transport. Therefore, the improvement in insulin resistance may also be related to the up-regulation of SGLT1 expression. Whether gut microbiota modulates the expression of nutrient-responsive receptors and transporters remains unclear. A previous study found up-regulation of SGLT1 and GLUT2 expression in mice lacking gut microbiota [38], implying that the microbiota could be a potential regulator of glucose homeostasis. This may be worth pursuing in future research.

The peptides in the gut are produced by specific enteroendocrine cells. The gut releases hormones to regulate glucose metabolism and balance glucose homeostasis when feeding. Our previous research indicated that high temperature significantly affects the levels of CCK, ghrelin, and GIP [8]. In the current study, the results showed that 0.4 mg/kg Cr as supplemented Cr-Pic decreased the levels of CCK in the jejunum and circulation, increasing the GIP level in the jejunum. CCK is an anorexigenic hormone and, in broilers, an important regulator of appetite inhibition that is involved in the neurohumoral regulation of appetite [39]. We previously mentioned that the feed intake of the 0.4 mg/kg Cr supplemented group significantly increased compared with that of the HT group, which means that the effect of appetite inhibition of CCK was weakened. GIP plays an important role in the regulation of blood glucose and insulin levels [40]. In addition, GIP could regulate glucose uptake and lipoprotein lipase lipolysis, which plays a prominent role in glucose metabolism in adipose tissue [41]. It was reported that GIP has been used as a novel hypoglycemic agent in diabetic patients [42]. Interestingly, the concentration of GIP in the jejunum was significantly increased in the current study. The rate of glucose absorption determines the amount of GIP [43]. In addition to playing a role in glucose transport, SGLT1 is also considered to be a glucose sensor in intestinal endocrine cells, promoting the secretion of GIP [36,44]. Hence, this may be due to up-regulated SGLT1 expression, because glucose-induced GIP secretion depends primarily on SGLT1 in the normal state [45]. In addition, several forms of evidence suggest the microbiota could promote or inhibit the secretion of gut peptides such as GIP and PYY by SCFAs [46,47]. However, the exact mechanisms through which CCK regulates glucose levels and appetite are not known, and its regulation may be similar to other gut peptides such as GIP. So far, there have been no relevant reports on this, and further research is needed. Nevertheless, as research on the brain–gut axis is still very limited, this study can provide a reference for the pathway and mechanisms of the brain–gut axis.

## 5. Conclusions

In conclusion, a diet with 0.4 mg/kg of supplemental Cr as Cr-Pic/kg improved performance, increased the expression of SGLT1, altered the structure of intestinal microbiota, affected the secretion of gastrointestinal peptides, and relieved insulin resistance in broilers under heat stress. Moreover, this was beneficial to maintain glucose homeostasis. Hence, a diet supplemented with 0.4 mg/kg Cr as Cr-Pic/kg could alleviate the damage caused by heat stress in broilers.

## Figures and Tables

**Figure 1 animals-12-00844-f001:**
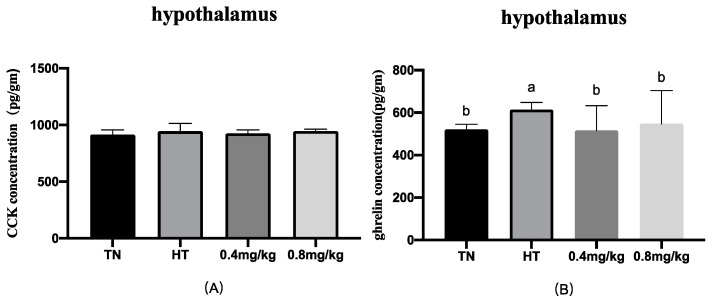
Effects of chromium supplementation on gastrointestinal peptides in the hypothalamus. (**A**) CCK concentration in hypothalamus; (**B**) ghrelin concentration in hypothalamus. ^a,b^ Means values between the column with different letters differ significantly.

**Figure 2 animals-12-00844-f002:**
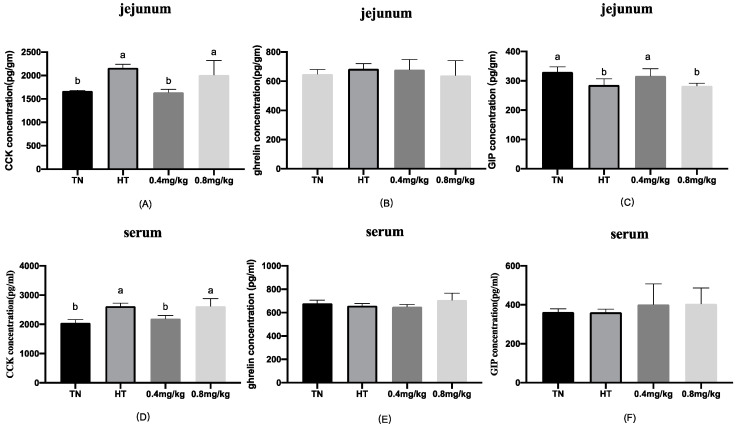
Effects of chromium supplementation on gastrointestinal peptides in the jejunum and serum. (**A**) CCK concentration in the jejunum; (**B**) ghrelin concentration in the jejunum; (**C**) GIP concentration in the jejunum; (**D**) CCK concentration in the serum; (**E**) ghrelin concentration in the serum; (**F**) GIP concentration in the serum. ^a,b^ Means values between the column with different letters differ significantly.

**Figure 3 animals-12-00844-f003:**
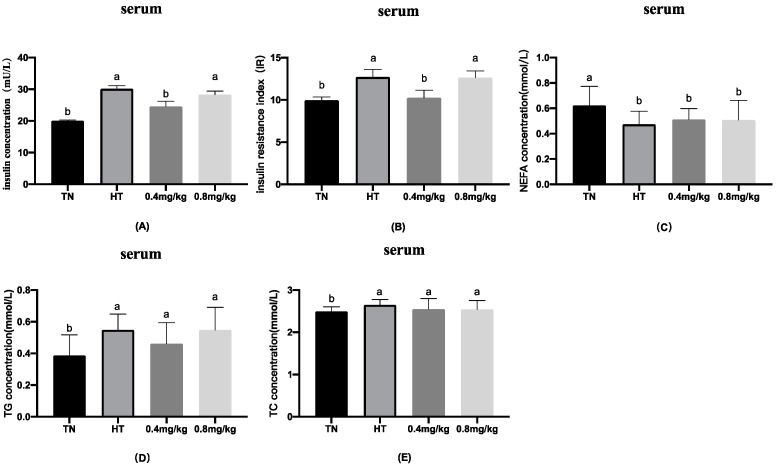
Effects of chromium supplementation on glucose homeostasis and other serum parameters. (**A**) Insulin concentration in the serum; (**B**) insulin resistance index in the serum; (**C**) NEFA concentration; (**D**) TG concentration in the serum; (**E**) TC concentration in the serum. ^a,b^ Means values between the column with different letters differ significantly.

**Table 1 animals-12-00844-t001:** List of abbreviations in this article.

Abbreviations	Full Name
AA	Arbor Acres
ADFI	average daily feed intake
ADG	average daily gain
CCK	cholecystokinin
Cr-pic	chromium picolinate
ELISA	enzyme-linked immunosorbent assay
FCR	feed conversion ratio
GIP	gastric inhibitory polypeptide
GLUT2	glucose transporter 2
HT	high-temperature
NEFA	non-esterified fatty acid
PBS	phosphate buffer saline
qPCR	quantitative real-time polymerase chain reaction
SGLT1	sodium-glucose transporter 1
TC	total cholesterol
TG	triglycerides
TN	thermal-neutral

**Table 2 animals-12-00844-t002:** Treatment group design.

	Adaptation Period	Temperature/ Humidity	Cr (+/−) ^1^
Thermal-neutral group (TN)	7 d	21 °C/60%	0 mg/kg
High-temperature group (HT)	7 d	31 °C/60%	0 mg/kg
0.4 mg/kg supplemented group	7 d	31 °C/60%	0.4 mg/kg
0.8 mg/kg supplemented group	7 d	31 °C/60%	0.8 mg/kg

^1^ Level of chromium supplemented.

**Table 3 animals-12-00844-t003:** Composition and nutrient levels of the basal diet.

Items	Content (%)	Items	Content (%)
Ingredients		Nutrient levels ^2^	
Corn	56.51	ME/(MJ/Kg)	12.73
Soybean meal	35.52	CP	20.07
Soybean oil	4.50	Ca	0.90
NaCl	0.30	AP	0.40
Limestone	1.00	Lys	1.00
CaHPO_4_	1.78	Met	0.42
d L-Met	0.11	Met + Cys	0.78
Premix ^1^	0.28		
Total	100.00		

^1^ Premix provided the following per kg of the diet: Vit A 10,000 IU, Vit D3 3400 IU, Vit E 16 IU, Vit K3 2.0 mg, Vit B1 2.0 mg, Vit B2 6.4 mg, Vit B6 2.0 mg, Vit B12 0.012 mg, pantothenic acid calcium 10 mg, nicotinic acid 26 mg, folic acid 1 mg, biotin 0.1 mg, choline 500 mg, Zn(ZnSO_4_·7H_2_O) 40 mg, Fe (FeSO_4_·7H_2_O) 80 mg, Cu(CuSO_4_·5H_2_O) 8 mg, Mn(MnSO_4_·H_2_O) 80 mg, I(KI) 0.35 mg, Se(Na_2_SeO_3_) 0.15 mg. ^2^ Calculated values.

**Table 4 animals-12-00844-t004:** Primer sequences.

Gene	Primer Name	Primer Sequences
SGLT1	SGLT1-1f	5-GGATCAACAATGCTGCGGAC-3
	SGLT1-1r	5-CACCTACTGTCCCTCGGTTG-3
GLUT2	GLUT2-2F	5-GAGAGCCCCCGCTATCTCTA-3
	GLUT2-2R	5-GCCTGAGAACTGCTGCGATA-3
YWHAZ (reference gene)	YWHAZ-F	5-TTGCTGCTGGAGATGACAAG-3
	YWHAZ-R	5-CTTCTTGATACGCCTGTTG-3

**Table 5 animals-12-00844-t005:** Effect of chromium supplementation on performance of heat-stressed broilers.

Treatments						
Items	TN	HT	HT + Cr 0.4 mg/kg	HT + Cr 0.8 mg/kg	SEM	*p*-Value
IABW (g)	1405.26	1410.58	1401.93	1408.07	17.25	>0.05
FABW (g)	2553.82 ^a^	2384.98 ^c^	2430.41 ^b^	2360.63 ^c^	23.37	<0.05
ADG (g/d)	82.04 ^a^	69.60 ^c^	72.82 ^b^	68.04 ^c^	0.65	<0.05
ADFI (g/d)	159.47 ^a^	143.41 ^c^	150.35 ^b^	142.93 ^c^	0.94	<0.05
FCR (g/g)	1.95 ^b^	2.06 ^a^	2.07 ^a^	2.10 ^a^	0.02	<0.05

Values are presented as means. SEM, standard error of the mean; TN, thermal neutral; HT, high ambient temperature; IABW, initial average body weight; FABW, final average body weight; ADG, average daily gain; ADFI, average daily feed intake; FCR, feed conversion rate. ^a,b,c^ Means values within the same line with different superscripts differ significantly. (*p* < 0.05).

**Table 6 animals-12-00844-t006:** Effect of chromium supplementation on cecum digesta microbiota composition under heat stress.

		Treatments				
Level	Species Name	TN	HT	HT + 0.4 mg/kg	SEM	*p*-Value
	*Actinobacteriota* (%)	3.78 ^a^	6.50 ^a^	0.73 ^b^	0.66	<0.05
Phylum	*Proteobacteria* (%)	0.08 ^a^	0.11 ^a^	0.01 ^b^	0.01	<0.05
	*Bacilli* (%)	6.32 ^b^	8.99 ^b^	20.60 ^a^	2.63	<0.05
Class	*Coriobacteriia* (%)	5.28 ^a^	6.46 ^a^	0.91 ^b^	0.65	<0.05
	*Gammaproteobacteria* (%)	0.05 ^a^	0.11 ^a^	0.01 ^b^	0.01	<0.05
Family	*Lactobacillaceae* (%)	4.01 ^b^	4.47 ^b^	8.64 ^a^	1.21	<0.05
	*Christensenellaceae* (%)	1.32 ^a^	0.46 ^b^	1.06 ^a^	0.11	<0.05
	*Erysipelotrichaceae* (%)	1.81 ^b^	3.63 ^b^	10.66 ^a^	1.23	<0.05
Genus	*Clostridiaceae* (%)	2.25 ^a^	2.95 ^a^	1.54 ^b^	0.27	<0.05
	*Ruminococcus* (%)	1.23 ^a^	1.00 ^a^	0.07 ^b^	0.03	<0.05
	*Turicibacter* (%)	0.98 ^b^	3.10 ^b^	8.20 ^a^	0.62	<0.05
	*Olsenella* (%)	3.53 ^a^	6.30 ^a^	0.01 ^b^	0.37	<0.05

Values are presented as means. SEM, standard error of the mean; TN, thermal neutral group; HT, high temperature group. ^a,b^ Means within the same line with different superscripts differ significantly. (*p* < 0.05).

**Table 7 animals-12-00844-t007:** Effects of chromium supplementation on SCFA concentrations.

		Treatments				
Item (μg/mL)	TN	HT	0.4 mg/kg	0.8 mg/kg	SEM	*p*-Value
acetate	543.95 ^b^	805.09 ^a^	752.15 ^a^	771.76 ^a^	23.02	>0.05
propionic acids	51.74	71.34	90.98	70.73	13.21	>0.05
butyric acids	254.71	236.88	227.07	261.29	21.99	>0.05

Values are presented as means. SEM, standard error of the mean; TN, thermal neutral group; HT, high temperature group. ^a,b^ Means within the same line with different superscripts differ significantly. (*p* < 0.05).

**Table 8 animals-12-00844-t008:** Effects of chromium picolinate supplementation on the level of glucose transporter expression.

Item	Treatments			SEM	*p*-Value
	TN	HT	0.4 mg/kg		
GLUT2 (liver)	1.21	0.94	0.91	0.51	>0.05
SGLT1 (jejunum)	3.18 ^a^	0.42 ^b^	1.29 ^a^	0.72	<0.05

Values are presented as means. SEM, standard error of the mean; TN, thermal neutral group; HT, high temperature group. ^a,b^ Means within the same line with different superscripts differ significantly. (*p* < 0.05).

## Data Availability

Not Applicable.

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
