# Peer review of "Effects of Dietary Chromium Picolinate on Gut Microbiota, Gastrointestinal Peptides, Glucose Homeostasis, and Performance of Heat-Stressed Broilers"

_animals, 2022, doi:10.3390/ani12070844_

Round 1
Reviewer 1 Report
The article is devoted to an urgent problem - the study of the negative impact of heat stress on the indicators of physiological stability, metabolic processes of industrial poultry - which directly affects productive indicators.
The authors reviewed the latest trends in the prevention of the effects of heat stress in animals, including the use of chromium preparations.
The manuscript has been prepared at a sufficiently high level. No major changes are required.
But there are some comments concerning the structure of the article:
- The authors use a significant number of abbreviations. Therefore, it is advisable to make a list of abbreviations used at the beginning of the article.
- In addition, it is not customary to use abbreviations in the abstract, this is quite inconvenient. Therefore, it is advisable to exclude all abbreviations from the abstract.
- Since animal experiments were conducted in the preparation of the manuscript, an Ethics Committee opinion must be provided.
- In conclusion, it is recommended to add perspectives and directions for further research in this area
Author Response
Dear reviewer,
Thank you for your letter and for the reviewer’s comments concerning our manuscript entitled “Effects of dietary chromium picolinate on gut-microbiota, gastrointestinal peptides, glucose homeostasis, and performance of heat-stressed broilers”(animals-163687). Those comments are valuable and very helpful for revising and improving our paper, as well as the important guiding significance to our research. We have studied comments carefully and have made corrections which we hope meet with approval. Revised portions are marked in red on the paper.
The main corrections in the paper and the responds to the reviewers comments are following:
1. The authors use a significant number of abbreviations. Therefore, it is advisable to make a list of abbreviations used at the beginning of the article.
Response: We have made a list of abbreviations behind the introduction, please see Table 1.
2. In addition, it is not customary to use abbreviations in the abstract, this is quite inconvenient. Therefore, it is advisable to exclude all abbreviations from the abstract
Response: We have excluded all abbreviations from the abstract.
3.Since animal experiments were conducted in the preparation of the manuscript, an Ethics Committee opinion must be provided.
Response: We have provided the ethic committee approval file to the "Animals", and we have descripted the ethic approval at the end of the article.
4.In conclusion, it is recommended to add perspectives and directions for further research in this area
Response: Thank you for your suggestion,and we have supplemented some perspectives in discussion section.
At last, we have revised the whole manuscript according to the problems you mentioned in the additional remarks.
We tried our best to improve the manuscript and made some changes in the manuscript. These changes will not influence the content and framework of the paper. And here we did not list the changes but marked in red in revised paper.
We appreciate for your warm work earnestly, and hope that the correction will meet with approval.
Once again, thank you very much for your comments and suggestion.
Reviewer 2 Report
The aim of the research was to determine the effect of dietary chromium picolinate on gut-microbiota, gastrointestinal peptides, glucose homeostasis, and performance of heat-stressed broiler chickens. The number broiler chickens used in the experiment is sufficient. The applied research methods are correct. The discussion is well conducted and comprehensive. Well-chosen references. Before publishing in Animals, the article requires additions and corrections. The proposed changes are listed below:
General comments:
Please prepare the article in accordance with the instructions for authors.
Please use the abbreviated name journal for item number 13, 19, 22, 27, 38, 40, 46 in References chapter
There must be „dot” after every parts abbrevaited name journal, for example, "Poult. Sci. " instead of Poult Sci
Detiled comments
L22 "three groups (HT)" instead of three group
L24 "HT and TN groups" instead of "HT group"
L22 after "31" a space
L24 CCK (full name) instead of current form
L25 GIP (full namre) instaed of current form
L28 Protejobacteria or Proteobacteria?
L31 Christensenellaceae - in relation to the NT group too?
L35 "... .decreased (p < 0.05)" - in relation to which groups?
L36 All parameters? But what parameters?
L79 + provide information about the type of building (closed, no windows?), type of floor/litter, pen dimensions, length, intensity, colour of light.
In table 1 there are spaces after the numbers when noting the temperature
L99, 102 between number and unit of measurement space, for example „2.0 mg” instead of 2.0mg
L109 Add information about the total number of birds selected for the study
L176 „… but was no significant change in the 0.8 mg / kg - is that so?
L178-179 only 0.8 mg / kg had no significant; 0.4 mg / kg is significant different
In table 4 - please add the data for BW 28 d and BW 42 d in grams
L184 "Values ​​are means ± SEM is misleading, SEM given for the entire population, not for individual data (groups)
L192 TN instead of HT
L198-201 "Compared with…. HT group (p < 0.05) - it was previously but spelled differently
L201-202 GIP in serum ghrelin in the hypothalamus serum - hardly understandable
L203-204 "The were no effects on ..." - is it so?
L221 Lactobacillaceae or Lachnospiraceae ?. Lactobacillaceae is higher in HT than in the NT group
L222 no data for Peptococcaceae in Table 5
L246-251 Description of the minors, no reference to which group
L259-262 Description not understandable - 2 were the 3 groups compared?
In tables 4-7, "p-Value" instead of "p value"
Author Response
Dear reviewer,
Thank you for your letter and for the reviewer’s comments concerning our manuscript entitled “Effects of dietary chromium picolinate on gut-microbiota, gastrointestinal peptides, glucose homeostasis, and performance of heat-stressed broilers”(animals-163687). Those comments are valuable and very helpful for revising and improving our paper, as well as the important guiding significance to our research. We have studied comments carefully and have made corrections which we hope meet with approval. Revised portions are marked in red on the paper.
The main corrections in the paper and the responses to the reviewer comments are following:
Please use the abbreviated name journal for item number 13, 19, 22, 27, 38, 40, 46 in References chapter
Response: We have changed the full name of the journal as the abbreviated form in the References chapter.
There must be „dot” after every parts abbreviated name journal, for example, "Poult. Sci. " instead of Poult Sci
Response: We have added all "dot" after the abbreviated journal name.
Detailed comments
As for the detailed comments, we tried our best to improve the manuscript and made some changes to the manuscript. These changes will not influence the content and framework of the paper. And here we did not list the changes but marked them in red in the revised paper.
L22 "three groups (HT)" instead of three group
Response: thanks, we have corrected it. please see line 22.
L24 "HT and TN groups" instead of "HT group"
Response: We have corrected that, please see line 24.
L22 after "31" a space
Response: We have added that. please see line22.
L24 CCK (full name) instead of the current form
Response: We have corrected that, please see line25.
L25 GIP (full namre) instaed of current form
Response: We have corrected that, please see line26.
L28 Protejobacteria or Proteobacteria?
Response: it is our mistake, and we have corrected that. please see line 29.
L31 Christensenellaceae - in relation to the NT group too?
Response: Christensenellaceae is of difference between HT group and TN group.
L35 "... .decreased (p < 0.05)" - in relation to which groups?
Response: We have added the group's expression, please see line 37.
L36 All parameters? But what parameters?
Response: We have added the parameters in the line 39.
L79 + provide information about the type of building (closed, no windows?), type of floor/litter, pen dimensions, length, intensity, colour of light.
Response: We have added the descriptions about the building and lighting program, please see line 91,99,100,104.
In table 1 there are spaces after the numbers when noting the temperature
Response: We have added the spaces in table 2 and whole manuscript.
L99, 102 between number and unit of measurement space, for example „2.0 mg” instead of 2.0mg
Response: We have added all the spaces in this part, please see line 110-112.
L109 Add information about the total number of birds selected for the study
Response: We have added that, please see line 120.
L176 „… but was no significant change in the 0.8 mg / kg - is that so?
Response: We have corrected that, please see line 187-189.
L178-179 only 0.8 mg / kg had no significant; 0.4 mg / kg is significant different
Response: We have corrected that. please see line 187-191.
In table 4 - please add the data for BW 28 d and BW 42 d in grams
Response: We have added the data, please see table 5.
L184 "Values ​​are means ± SEM is misleading, SEM given for the entire population, not for individual data (groups)
Response: We have corrected that and made some adjustments. please see line 208,295,344.
L192 TN instead of HT
Response: Thanks, we have corrected that.please see line 217.
L198-201 "Compared with…. HT group (p < 0.05) - it was previously but spelled differently
Response: We have made some changes about this part.
L201-202 GIP in serum ghrelin in the hypothalamus serum - hardly understandable
Response: We have changed the expressions, please see line 225.
L203-204 "The were no effects on ..." - is it so?
Response: We have made some adjustments, please see line 225-226
L221 Lactobacillaceae or Lachnospiraceae ?. Lactobacillaceae is higher in HT than in the NT group
Response: We have corrected this part, please see line 273.
L222 no data for Peptococcaceae in Table 5
Response: it is a mistake we made, and we have excluded this bacterial, please see line 270.
L246-251 Description of the minors, no reference to which group
Response: we have added the reference in this part, please see line 314-319
L259-262 Description not understandable - 2 were the 3 groups compared?
Response: we have made some adjustments to the expressions, please see line 327-331
In tables 4-7, "p-Value" instead of "p value"
Response: We have corrected that in all tables.
At last, we have revised the whole manuscript according to the problems you mentioned in the additional remarks.
We appreciate your warm work earnestly and hope that the correction will meet with approval
Once again, thank you very much for your comments and suggestion.
Reviewer 3 Report
Lines 83-84 “Then 220 chickens were randomly allocated to four groups with five replicates (11 chickens/pen) according to individual body weight “modify; you must indicate the beginning and end of the experiment and the beginning of the adaptation period.
Table 2 – “Premix provided the following per kg of the diet: VA 10 000 IU, VD3 3 400 IU, VE 16 IU, VK3 2.0 mg, VB1 2.0mg, VB2 6.4mg, VB6 2.0mg, VB12 0.012mg” change V as vit.
Lines 175-178 change “The ADFI of the 0.4 mg/kg group was significantly higher than that of the HT group (p < 0.05), but there was no significant change in the 0.8 mg/kg group, the ADFI of 0.4 mg/kg and 0.8 mg/kg groups were significantly lower than that of the TN group (p < 0.05).” as “The ADFI of the TN group was significantly higher than those of the HT, 0,4 mg/kg and 0,8 mg/kg groups (p < 0.05), the ADFI of 0.4 mg/kg group was significantly higher than those of the HT and 0,8 mg/kg groups (p < 0.05).”
Lines 178-179 change “High ambient temperature decreased the ADG (p < 0.05), while the ADG of 0.4 mg/kg and 0.8 mg/kg groups had no significant change compared with the HT group” as “High ambient temperature decreased the ADG (p < 0.05), while the ADG of 0.4 mg/kg was significantly higher than those of the HT and 0.8 mg/kg groups”
Line 192 change “supplementation groups and the HT group” as “supplementation groups and the TN group”
Lines 192-194 change “In jejunum and serum, the concentration of CCK in jejunum and serum of HT group was higher than that in the TN group (p < 0.05), while the CCK in the 0.4 mg/kg group was significantly lower than that in the HT group” as “ the concentration of CCK in jejunum and serum of HT group was higher than that in the TN group (p < 0.05), while the CCK in the 0.4 mg/kg group was significantly lower than those in the HT and 0,8 mg/kg groups”
Lines 196-198 change “In jejunum and serum, compared with the TN group, the GIP content in the HT group was lower, while the GIP content in the 0.4 mg/kg group was significantly higher than that in the HT group” as “In jejunum, compared with the TN group, the GIP content in the HT group was lower, while the GIP content in the 0.4 mg/kg group was significantly higher than those in the HT and 0,8 mg/kg groups”
Lines 198-199 delete “Compared with the HT group, GIP concentration in the jejunum of 0.4 mg/kg supplemented group was significantly increased (p < 0.05)”
Lines 201-204 change “The CCK in the hypothalamus, GIP in serum ghrelin in the hypothalamus serum, and jejunum in the 0.4 mg/kg supplemented group was not significantly different from that of the HT group. There were no effects on all examined gastrointestinal peptides in the 0.8 mg/kg supplemented group.” As “The CCK in the hypothalamus, GIP in serum, ghrelin in serum and jejunum in the 0.4 mg/kg supplemented group was not significantly different from that of the HT group.”
Lines 216-218 change “At the phylum level, the proportion of Actinobacteriota and Proteobacteria of 0.4 mg/kg supplemented group decreased compared to the HT group (p < 0.05)” as “At the phylum level, the proportion of Actinobacteriota and Proteobacteria of 0.4 mg/kg supplemented group decreased compared to the HT and TN groups (p < 0.05)”
Lines 218-220 change “At the class level, the proportion of Bacilli of 0.4 mg/kg supplemented group was significantly increased, whereas Coriobacteriia and Gammaproteobacteria were significantly decreased compared with the HT group (p < 0.05).” as “At the class level, the proportion of Bacilli of 0.4 mg/kg supplemented group was significantly increased, whereas Coriobacteriia and Gammaproteobacteria were significantly decreased compared with the HT and TN groups (p < 0.05).”
Lines 221-223 “the proportion of Lachnospiraceae and Christensenellaceae of HT group has decreased significantly compared with the TN group, whereas the proportion of Peptococcaceae of HT group decreased compared with the TN group” Non true for Lactobacillaceae; Peptococcaceae ?
Line 246 change “and TC of the HT group increased, and the level of NEFA of the HT group decreased” as “and TC of the HT group increased, and the level of NEFA decreased”
Lines 259-260 change “Compared with the TN group, the expression of the SGLT1 gene was decreased.” As “Compared with the TN group, the expression of the SGLT1 gene was decreased in the HT group.”
Line 276 after “broilers under heat stress [25]” please add also 26.
26 – Untea, A.E.; Varzaru, J.; Turcu, R.P.; Panaite, T.D.; Saracila M. The use of dietary chromium associated with vitamins and minerals (synthetic and natural source) to improve some quality aspects of broiler thigh meat reared under heat stress condition. It. J. Anim. Sci. 2021, 20, 1491–1499. https://doi.org/10.1080/1828051X.2021.1978335
Lines 394-395 “Sun, X.; Zhang, H.; Sheikhahmadi, A.; Wang, Y.; Jiao, H.; Lin, H.; Song, Z. Effects of Heat Stress on the Gene Expression of 394 Nutrient Transporters in the Jejunum of Broiler Chickens (Gallus. Int J Biometeorol 2014.” Add pages.
Author Response
Dear reviewer,
Thank you for your letter and for the reviewer’s comments concerning our manuscript entitled “Effects of dietary chromium picolinate on gut-microbiota, gastrointestinal peptides, glucose homeostasis, and performance of heat-stressed broilers”(animals-163687). Those comments are valuable and very helpful for revising and improving our paper, as well as the important guiding significance to our research. We have studied comments carefully and have made corrections which we hope meet with approval. Revised portions are marked in red on the paper.
The main corrections in the paper and the responses to the reviewer comments are following:
Lines 83-84 “Then 220 chickens were randomly allocated to four groups with five replicates (11 chickens/pen) according to individual body weight “modify; you must indicate the beginning and end of the experiment and the beginning of the adaptation period.
Response: We have added the details about the trial arrangement, please see line 96-97
Table 2 – “Premix provided the following per kg of the diet: VA 10 000 IU, VD3 3 400 IU, VE 16 IU, VK3 2.0 mg, VB1 2.0mg, VB2 6.4mg, VB6 2.0mg, VB12 0.012mg” change V as vit.
Response: we have made some adjusting, please see line 110-111.
Lines 175-178 change “The ADFI of the 0.4 mg/kg group was significantly higher than that of the HT group (p < 0.05), but there was no significant change in the 0.8 mg/kg group, the ADFI of 0.4 mg/kg and 0.8 mg/kg groups were significantly lower than that of the TN group (p < 0.05).” as “The ADFI of the TN group was significantly higher than those of the HT, 0,4 mg/kg and 0,8 mg/kg groups (p < 0.05), the ADFI of 0.4 mg/kg group was significantly higher than those of the HT and 0,8 mg/kg groups (p < 0.05).”
Lines 178-179 change “High ambient temperature decreased the ADG (p < 0.05), while the ADG of 0.4 mg/kg and 0.8 mg/kg groups had no significant change compared with the HT group” as “High ambient temperature decreased the ADG (p < 0.05), while the ADG of 0.4 mg/kg was significantly higher than those of the HT and 0.8 mg/kg groups”
Line 192 change “supplementation groups and the HT group” as “supplementation groups and the TN group”
Lines 192-194 change “In jejunum and serum, the concentration of CCK in jejunum and serum of HT group was higher than that in the TN group (p < 0.05), while the CCK in the 0.4 mg/kg group was significantly lower than that in the HT group” as “ the concentration of CCK in jejunum and serum of HT group was higher than that in the TN group (p < 0.05), while the CCK in the 0.4 mg/kg group was significantly lower than those in the HT and 0,8 mg/kg groups”
Lines 196-198 change “In jejunum and serum, compared with the TN group, the GIP content in the HT group was lower, while the GIP content in the 0.4 mg/kg group was significantly higher than that in the HT group” as “In jejunum, compared with the TN group, the GIP content in the HT group was lower, while the GIP content in the 0.4 mg/kg group was significantly higher than those in the HT and 0,8 mg/kg groups”
Lines 198-199 delete “Compared with the HT group, GIP concentration in the jejunum of 0.4 mg/kg supplemented group was significantly increased (p < 0.05)”
Lines 201-204 change “The CCK in the hypothalamus, GIP in serum ghrelin in the hypothalamus serum, and jejunum in the 0.4 mg/kg supplemented group was not significantly different from that of the HT group. There were no effects on all examined gastrointestinal peptides in the 0.8 mg/kg supplemented group.” As “The CCK in the hypothalamus, GIP in serum, ghrelin in serum and jejunum in the 0.4 mg/kg supplemented group was not significantly different from that of the HT group.”
Lines 216-218 change “At the phylum level, the proportion of Actinobacteriota and Proteobacteria of 0.4 mg/kg supplemented group decreased compared to the HT group (p < 0.05)” as “At the phylum level, the proportion of Actinobacteriota and Proteobacteria of 0.4 mg/kg supplemented group decreased compared to the HT and TN groups (p < 0.05)”
Lines 218-220 change “At the class level, the proportion of Bacilli of 0.4 mg/kg supplemented group was significantly increased, whereas Coriobacteriia and Gammaproteobacteria were significantly decreased compared with the HT group (p < 0.05).” as “At the class level, the proportion of Bacilli of 0.4 mg/kg supplemented group was significantly increased, whereas Coriobacteriia and Gammaproteobacteria were significantly decreased compared with the HT and TN groups (p < 0.05).”
Lines 259-260 change “Compared with the TN group, the expression of the SGLT1 gene was decreased.” As “Compared with the TN group, the expression of the SGLT1 gene was decreased in the HT group.”
Response: We very much appreciate your revision of our manuscript, our expression about the results is not appropriate. Therefore, we have changed the expression in accord with your advice.
Lines 221-223 “the proportion of Lachnospiraceae and Christensenellaceae of HT group has decreased significantly compared with the TN group, whereas the proportion of Peptococcaceae of HT group decreased compared with the TN group” Non true for Lactobacillaceae; Peptococcaceae ?
Response: we have made a mistake in this part, and we have corrected the expressions, please see line 265-269
Line 276 after “broilers under heat stress [25]” please add also 26.
Response: We have added this quote to the manuscript, please see line 355.
Lines 394-395 “Sun, X.; Zhang, H.; Sheikhahmadi, A.; Wang, Y.; Jiao, H.; Lin, H.; Song, Z. Effects of Heat Stress on the Gene Expression of 394 Nutrient Transporters in the Jejunum of Broiler Chickens (Gallus. Int J Biometeorol 2014.” Add pages.
Response: We have added pages in this part, please see line 482
At last, we have revised the whole manuscript according to the problems you mentioned in the additional remarks.
We tried our best to improve the manuscript and made some changes in the manuscript. These changes will not influence the content and framework of the paper. And here we did not list the changes but marked them in red in the revised paper.
We appreciate your warm work earnestly and hope that the correction will meet with approval
Once again, thank you very much for your comments and suggestion.
Round 2
Reviewer 3 Report
The authors corrected the work as suggested by the reviewers. The manuscript has been sufficiently improved and can now be published in its current form.